# A magnetic hydrogel for the efficient retrieval of kidney stone fragments during ureteroscopy

T. Jessie Ge[1,2], Daniel Massana Roquero[1,2], Grace H. Holton[1], Kathleen E. Mach[1,2], Kris Prado[1], Hubert Lau[2,3], Kristin Jensen[2,3], Timothy C. Chang[1,2], Simon Conti[1], Kunj Sheth[1], Shan X. Wang [ID][4] & Joseph C. Liao [ID][1,2] ✉

Only 60-75% of conventional kidney stone surgeries achieve complete stone-free status. Up to 30% of patients with residual fragments <2 mm in size experience subsequent stone-related complications. Here we demonstrate a stone retrieval technology in which fragments are rendered magnetizable with a magnetic hydrogel so that they can be easily retrieved with a simple magnetic tool. The magnetic hydrogel facilitates robust in vitro capture of stone fragments of clinically relevant sizes and compositions. The hydrogel components exhibit no cytotoxicity in cell culture and only superficial effects on ex vivo human urothelium and in vivo mouse bladders. Furthermore, the hydrogel demonstrates antimicrobial activity against common uropathogens on par with that of common antibiotics. By enabling the efficient retrieval of kidney stone fragments, our method can lead to improved stone-free rates and patient outcomes.

Kidney stones affect about 1 in 9 people and can cause significant morbidity ranging from severe pain to life-threatening infection and kidney damage[1]. Kidney stone disease represents a significant healthcare burden, leading to >1.3 million emergency room visits and >$5 billion in healthcare expenditures per year in the United States[2,3]. Ureteroscopic laser lithotripsy, an endoscopic surgical procedure, is the most common treatment and uses a laser fiber to fragment stones into smaller pieces. The fragments can then be retrieved or left to pass. Rendering a patient stone-free is the best way to prevent further complications or repeat interventions due to residual stone fragments, yet reported stone-free rates from ureteroscopic approaches are only 60–75%[4]. Small fragments and 'dust' (e.g. fragments less than 2 mm in diameter) that are difficult or impossible to retrieve with conventional stone extraction tools are commonly left behind under the assumption that they will spontaneously pass in the urine. However, many of these small stone fragments do not always pass due to the patient's anatomy and physical mobility. In addition, the fragmentation of large stones can generate a large volume of dust that impairs visualization of

remaining stone, leading to incomplete fragmentation and stone retrieval. Residual fragments can lead to recurrent symptoms and serve as a nidus for larger stone formation[5].

Active retrieval of stone fragments is currently largely performed with wire baskets, which are introduced through a ureteroscope and carefully manipulated to enclose around a stone and retrieve it from the body. Small fragments within the complex honeycomb-like anatomy can be difficult or impossible to grasp with the basket, despite the variable basket configurations on the market. Other technologies have been explored to improve stone fragment clearance. Focused ultrasound has been used to propel and reposition residual stone fragments in the postoperative setting[6]. Intraoperatively, some approaches focus on adhering small fragments together with autologous blood clots[7] or biopolymers[8,9] to facilitate retrieval with a basket. Negative pressure aspiration is an attractive tool to suction out stone fragments during surgery, but is limited by the small working channel of standard ureteroscopes to maintain patency, flow and visibility. Ureteroscopic aspiration systems have utilized catheters guided

[1]Department of Urology, Stanford University, Stanford, CA 94305, USA. [2]Veterans Affairs Palo Alto Health Care System, Palo Alto, CA 94304, USA. [3]Department of Pathology, Stanford University, Stanford, CA 94305, USA. [4]Department of Materials Science and Engineering, Stanford University, Stanford, CA 94305, USA. ✉e-mail: jliao@stanford.edu

by fluoroscopy rather than direct vision in order to increase the effective flow rate[10]. Previous attempts at magnetizing kidney stones and retrieving them with a magnetic tool have been limited by low magnetic strengths[11].

Here we present MagSToNE (Magnetic System for Total Nephrolith Extraction) to improve the efficiency of stone fragment clearance. Kidney stones are coated with a magnetic hydrogel and retrieved by a magnetic guidewire with alternating polarities to maximize the attractive magnetic force (Fig. 1). The hydrogel is composed of superparamagnetic iron oxide nanoparticles (SPIONs) which bind to exposed calcium ions on the stone surface, and a biopolymer (chitosan) which aggregates the nanoparticles on the stone via ionic gelation. The system is designed to be compatible with standard ureteroscopes in clinical use. We characterize the performance of this system in vitro with patient-derived kidney stone fragments. We further examine the biocompatibility of the hydrogel components and their effect on the urothelium and demonstrate the antimicrobial properties of the hydrogel against uropathogens commonly present in stone-forming patients.

## Results

### MagSToNE design

The majority of human kidney stones contain calcium. Approximately 70–80% of stones are composed of majority calcium oxalate monohydrate (COM) or dihydrate (COD), an additional 10–20% majority calcium phosphate (CaP), and a minority are purely uric acid or struvite (magnesium ammonium phosphate), and cystine[12]. Kidney stones are crystalline structures, and cations on the crystal face can ionically interact with and adhere to anionic functional groups. Carboxylic acid groups have demonstrated high adhesion forces to kidney stone crystal surfaces[13] and have been shown to inhibit stone crystal growth by binding to the exposed calcium or magnesium ions[14]. MagSToNE utilizes SPIONs functionalized with carboxylic acid-containing molecules to bind to and coat kidney stone fragments, thus rendering the stone fragments magnetizable. In the presence of an external magnetic field generated by the magnetic wire, a stone fragment experiences an attractive magnetic force

equal to $n \times F_m$, where $n$ is the number of superparamagnetic particles bound to the fragment, and the magnetophoretic force $F_m$ on a particle in a magnetic field is described by:[15]

$$F_m = \frac{V\chi}{\mu_0}\left(\vec{B}\cdot\nabla\right)\vec{B} \tag{1}$$

where $V$ is the volume of the particle, $\chi$ is the magnetic susceptibility of the particle (assuming the magnetic susceptibility of the surrounding buffer is zero), $\mu_0 = 4\pi\times10^{-7}(TmA^{-1})$, and $\mathbf{B}$ is the magnetic flux density ($T$).

The magnetophoretic force is proportional to the gradient of the magnetic field $\nabla B$. We have designed the magnetic wire to generate high magnetic gradients ($10^3$–$10^4$ T/m) (Supplementary Fig. S1) along the length of the wire by alternating diametric magnetic polarities within the wire. In contrast, a magnetic wire unidirectionally magnetized along its axis only generates high magnetic gradients and forces at the tip (Fig. 2a). Due to size constraints in endoscopic surgery, instruments and wires must generally be smaller than 3.6 French (Fr, or 1.2 mm in diameter) to fit through the working channel of a ureteroscope, but the scope and instrument can be withdrawn *en bloc* through a ureteral access sheath with diameter ranging from 10–12 Fr (3.3–4 mm in diameter). Thus, a diametrically magnetized wire that captures stones across its entire length rather than only at its tip can retrieve a much larger stone burden on a single pass compared to an axially magnetized wire, (Fig. 2a).

The alternating magnetic polarities create a metastable arrangement, and the individual magnets must be fixed or constrained. The cylindrical magnets are sheathed within a biocompatible plastic tubing, but the presence of the sheath limits the proximity of a stone fragment to the magnet and therefore limits the maximum magnetic gradient that it can encounter (Fig. 2b). Thinner sheaths achieve superior stone capture (Fig. 2c).

### Magnetic capture of kidney stone fragments in vitro

We initially tested magnetic kidney stone capture with the magnetic wire and carboxylic acid-coated superparamagnetic particles alone.

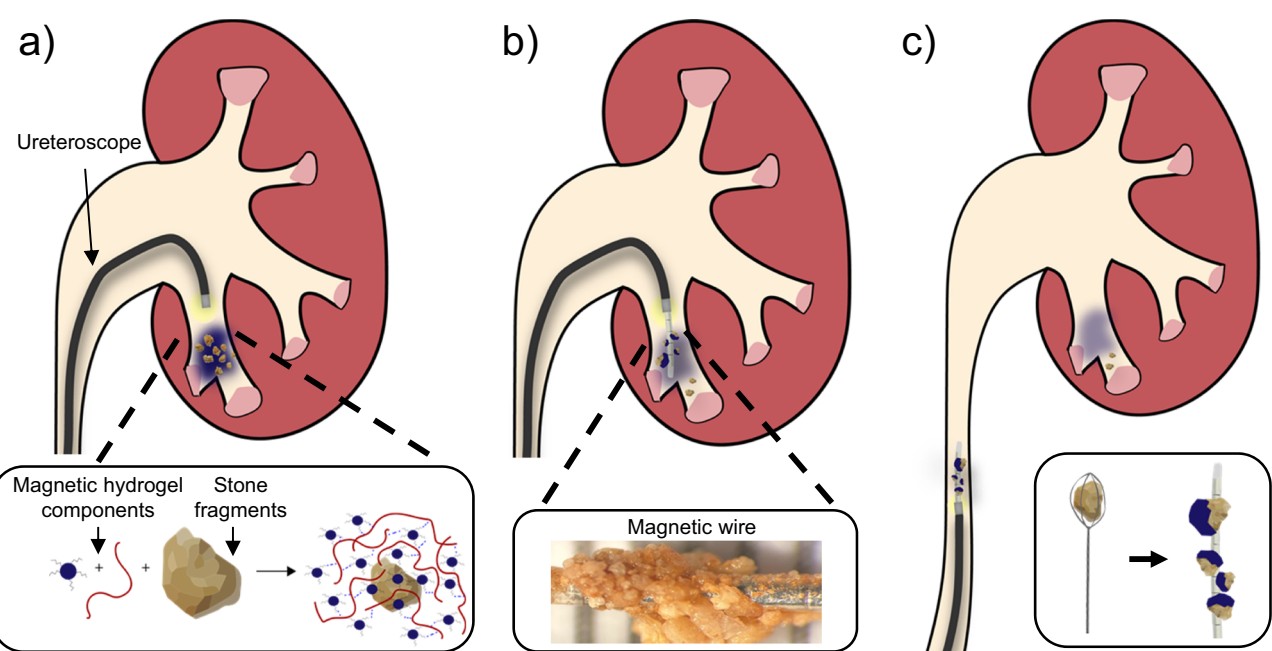

**Fig. 1 | MagSToNE schematic. a** SPIONs (blue circle) and a hydrogel-forming biopolymer (red line) are introduced through the working channel of a ureteroscope to coat stone fragments in a superparamagnetic hydrogel. **b** A magnetic wire is introduced to capture the magnetically labeled stone fragments. **c** The ureteroscope and wire, along with the captured fragments, are removed from the body. Inset compares capture of single fragments with a standard basket (left) to capture of multiple fragments with the magnetic wire.

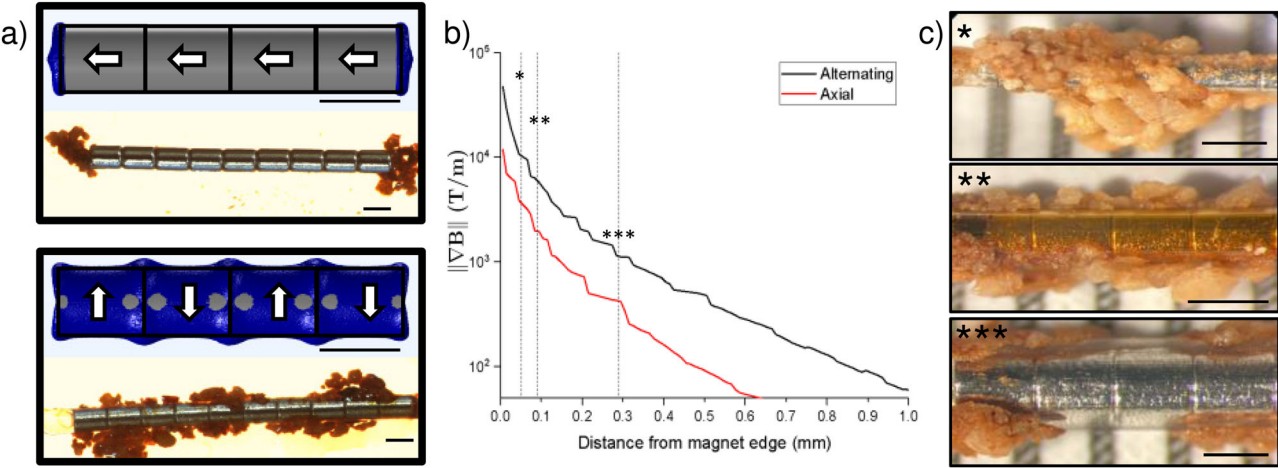

**Fig. 2 | Design and magnetic gradient of magnetic wire. a** Simulated magnetic gradient norm isosurface of 2000 T/m (blue) and actual image of magnetic stone capture by magnets with unidirectional axial polarities (top) versus alternating diametric polarities (bottom). White arrows point to the north pole of the magnet. Black lines indicate a scale of 1 mm. **b** Magnetic gradient radial decay for axial (red) versus alternating (black) magnetic wires. Dotted lines and asterisks indicate the sheath thicknesses depicted in (**c**). **c** Stone capture by magnetic wires with corresponding sheath thicknesses. Thicker sheaths are associated with lower surface gradients and lower capture efficiencies. (* sheath thickness = 0.05 mm, gradient at surface = 10758 T/m, capture efficiency = 91%; ** sheath thickness = 0.09 mm, gradient at surface = 6374 T/m, capture efficiency = 84%; *** sheath thickness = 0.29 mm, gradient at surface = 1151 T/m, capture efficiency = 18%). Source data are provided as a Source Data file.

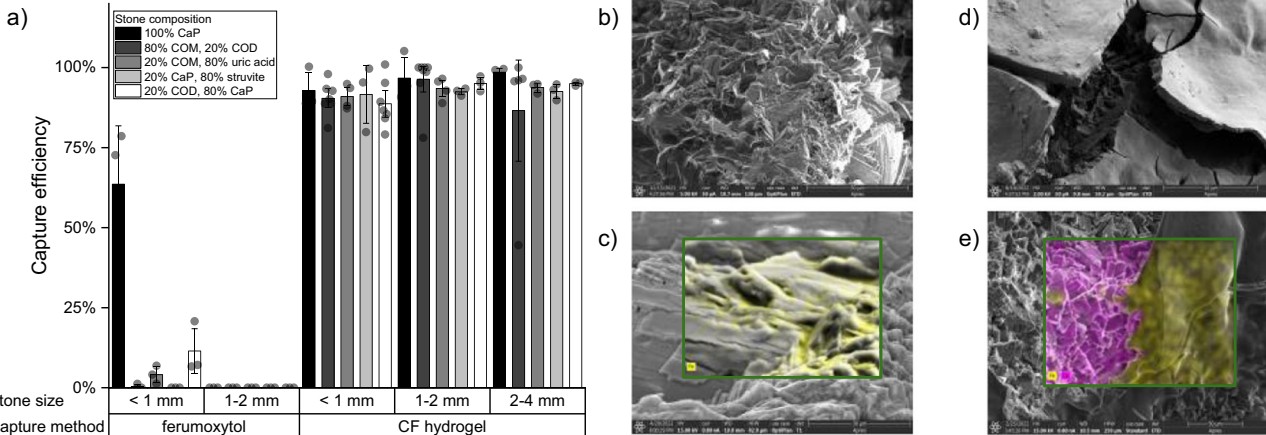

**Fig. 3 | In vitro performance of MagSToNE with and without CF hydrogel. a** Capture efficiency of ferumoxytol alone versus CF hydrogel for stones of different compositions and sizes. N = 3-8 independent experiments for each group. Data represents mean ± 1.5 standard error. Source data are provided as a Source Data file. **b**−**e** SEM images of 80% COM 20% COD stone. **b** Uncoated stone. **c** Stone coated with ferumoxytol alone. Inset shows energy-dispersive X-ray (EDX) signal for iron (yellow) corresponding to ferumoxytol on the surface of the stone. **d** Stone coated with CF hydrogel. The microns thick layer of hydrogel is cracked and reveals normal stone structure below. **e** Stone coated with CF hydrogel. Inset shows EDX signal for iron (yellow) and calcium (pink), corresponding to areas of CF hydrogel-coated stone and uncoated stone, respectively.

While large (50 nm to 3 μm) particles bound to the stone fragment surface (Supplementary Fig. S2) and reliably captured small (<1 mm) stone fragments of varying compositions, they were generally unable to facilitate capture of larger fragments. To successfully capture a stone, the magnetophoretic force must exceed the countering forces of gravity and drag, as well as surface tension should the magnetically captured stone encounter an air-fluid interface (i.e. bubble). The gravitational force on a stone increases with the cube of its diameter, while the magnetophoretic force is directly proportional to the number of particles bound to the stone, which for a monolayer surface coating of particles only increases with the square of the stone diameter. Thus, to capture larger stones, we utilized a hydrogel of superparamagnetic nanoparticles to bind a thick, multi-layer magnetic coating to the stone surface and significantly increase the number of particles per stone.

We developed a hydrogel of ferumoxytol, a 7 nm diameter SPION with a negatively charged carboxydextran coating, and chitosan, a cationic biopolymer. Despite the higher capture shown by larger magnetic particles, we decided to employ the 7 nm ferumoxytol nanoparticles due to their larger surface area to volume ratio which is beneficial for polymer cross-linking. We compared the capture efficiencies of MagSToNE using ferumoxytol alone versus the chitosan-ferumoxytol (CF) hydrogel for stones of different sizes and compositions (Fig. 3). While ferumoxytol alone bound to stones due to the carboxylic acid groups in its surface coating and captured about 60% of small (<1 mm diameter) calcium phosphate fragments in vitro, it performed inconsistently for stones of other compositions and could not capture larger fragments. In contrast, the CF hydrogel captured 100% of stones for all compositions tested, including larger fragments up to 4 mm in diameter and stones with as little as 20% calcium

content. These in vitro experiments are corroborated by particle tracing simulations that demonstrate that small stones (e.g. 1 and 3 mm) can be captured from several millimeters away (Supplementary Movies S1 and S2). Scanning electron microscopy (SEM) of magnetically labeled calcium oxalate stone fragments confirmed the binding of ferumoxytol and CF hydrogel to the stone surface (Fig. 3b–e). The small size of ferumoxytol precluded its reliable resolution on SEM, thus energy-dispersive X-ray (EDX) techniques were used to demonstrate the presence of iron on the stone surface. The CF hydrogel was easily seen, forming layers tens of microns thick. The lyophilized hydrogel was characterized by SEM and FT-IR spectroscopy (Supplementary Fig. S3). The highly heterogeneous and porous structure is characteristic of ionic gelation in which the water-soluble polymer is quickly cross-linked resulting in an insoluble polymeric nanocomposite.

### Cytotoxicity of magnetic hydrogel components

We next investigated the potential cytotoxic effects of the CF hydrogel and its components. The components were incubated with urothelial cell culture for up to 4 h, with final concentrations of 1.5 mg/mL ferumoxytol and 0.05% w/v chitosan representing tenfold dilutions of the stock solutions which would be used in the MagSToNE application. In clinical practice, urothelial cells will be exposed to high concentrations of the components for only seconds to minutes, as irrigation is constantly used during surgery and natural urine production will also wash away the components. The individual components and resultant CF hydrogel demonstrated no significant cytotoxicity in cell culture (Fig. 4A).

Chitosan has known time- and concentration-dependent exfoliative effects on murine and porcine urothelium related to its electrostatic effect with negatively charged integrins on the cell surface[16], with applications in drug delivery[17] and treatment of urinary tract infections[18]. Notably, previous studies have shown rapid and complete functional recovery of the urothelium after chitosan exposure[19–21]. Recognizing that monolayer cell culture may not adequately represent interactions between chitosan and the multi-layered urothelium, we performed additional studies on ex vivo human and in vivo murine urothelium. Fresh human urothelium was obtained from nephrectomy specimens and incubated with stock and diluted (1/10×) concentrations of ferumoxytol, chitosan, and CF hydrogel for representative periods of time. Prussian blue staining did not demonstrate any retained iron particles (Supplementary Fig. S4). Chitosan alone had an expected time and dose-dependent exfoliative effect on the urothelium, without apparent effect on submucosal tissues. Urothelium exposed to CF hydrogel for 1 min exhibited minimal changes, and urothelium exposed to CF hydrogel for 30 min showed exfoliation of only superficial layers of urothelial cells compared to the more significant exfoliation seen with exposure to chitosan (Fig. 4B).

We next studied the effects in the live murine bladder, as ex vivo studies are not well-suited to assess for an inflammatory response or regeneration. Mouse bladders were exposed to chitosan or CF hydrogel for 15 min. The mice were sacrificed at either 30 min or 12 h post exposure to examine the bladder urothelium. While there was significant exfoliation seen immediately following chitosan exposure, there was regeneration of the basal cell layer by 12 h. The CF hydrogel showed markedly reduced exfoliation of the urothelium compared to chitosan alone immediately following exposure, and the urothelium had returned to normal by 12 h post CF hydrogel exposure (Fig. 4C).

### Antimicrobial properties of ferumoxytol and CF hydrogel

The well-known antimicrobial properties of chitosan, similarly related to electrostatic interactions between positively charged chitosan and negatively charged macromolecules on bacterial cell surfaces leading to changes in cell permeability[22], motivated us to study the antibacterial effect of the CF hydrogel against common uropathogens. *Escherichia coli* (*E. coli*) and *Proteus mirabilis* (*P. mirabilis*) are the most common uropathogens isolated from patients with nephrolithiasis[23]. We analyzed the growth of ciprofloxacin sensitive bacterial clinical isolates exposed to individual components of the hydrogel over a 4-h period. (0.05% chitosan solution and 3 mg/ml ferumoxytol), the CF hydrogel, a no treatment control, and a positive control (4 μg/ml ciprofloxacin). Optical density measurements at 600 nm (OD600) were taken every 30 min over a 4-h period to quantify bacterial growth. In the *E. coli* group, both 0.05% chitosan and dilute CF hydrogel completely inhibited bacterial growth, similar to the antibiotic ciprofloxacin. For *P. mirabilis*, chitosan immediately inhibited bacterial growth, while both ciprofloxacin and the CF hydrogel showed inhibition of bacterial growth after 90–120 min. 3 mg/mL ferumoxytol alone neither inhibited nor promoted bacterial growth of either pathogen (Fig. 5). The delayed response of *P. mirabilis* may be related to bacterial adaptive responses such as swarming leading to decreased susceptibility to antibiotics[24,25]. The difference between chitosan and the CF hydrogel could be related to the reduction of free positively charged amino groups in the cross-linked chitosan. Growth inhibition was further confirmed by serial dilution and plating to verify that our optical density-derived growth curves were measuring viable cells, and no bacterial colonies were seen on plates derived from the chitosan and CF hydrogel groups for either pathogen (Supplementary Fig. S5).

## Discussion

Ureteroscopic clearance of kidney stone fragments can be tedious and incomplete, resulting in residual stone fragments that can lead to further patient morbidity including renal colic and secondary stones. We have demonstrated robust and reliable retrieval of kidney stone fragments with MagSToNE, using a superparamagnetic CF hydrogel and magnetic wire. The stones are first coated with a layer of SPIONs which nucleate the formation of a thick layer of SPION-containing hydrogel onto the stone surface. This drastically increases the number of bound SPIONs compared to a monolayer of bound SPIONs in the absence of chitosan, and the magnetic moment of the stone, which is necessary to magnetically capture larger and heavier stones. MagSToNE additionally captures stone fragments of varying chemical compositions, improving its versatility in clinical use. While the carboxylic acid functional groups on the SPIONs are intended to bind to exposed calcium ions on the stone surface, even stones with as little as 20% calcium content were reliably captured. Carboxylic acid groups likely also bind to exposed magnesium ions on struvite stones[14].

While chitosan is widely regarded as a biodegradable and biocompatible nontoxic polymer commonly used in biomedical hydrogels, it has known exfoliative effects on the urothelium[16]. The CF hydrogel and its components (chitosan and ferumoxytol) demonstrated no statistically significant cytotoxicity in human urothelial cell culture. At the exposure times expected in clinical practice (on the order of 1 min due to the constant use of irrigation during ureteroscopy), we did not see significant damage to the urothelium in ex vivo human urothelium. Even at 30 min, the CF hydrogel only showed exfoliation of a few cell layers. In in vivo murine bladders, urothelium recovered quickly from the exfoliative effects of chitosan, and these effects were further mitigated by the hydrogel formulation. Furthermore, the SPION used in our hydrogel, ferumoxytol, is FDA-approved for the treatment of iron deficiency anemia and has demonstrated safety data for intravenous use. We anticipate that use in the urinary tract for limited amounts of time will further reduce any potential risk. Prussian blue staining for iron showed that CF hydrogel was easily washed away from human urothelium after ex vivo incubation. In mouse experiments, the mice were seen to void most of the excess CF hydrogel from their bladder. Small amounts of CF hydrogel can be retained in the bladder after the first void but appear to be cleared with subsequent voids (Supplementary Fig. S4c). In clinical practice, the majority of excess CF hydrogel will either be irrigated away, or easily retrieved with the magnetic wire. In our in vitro experiments, 90% of

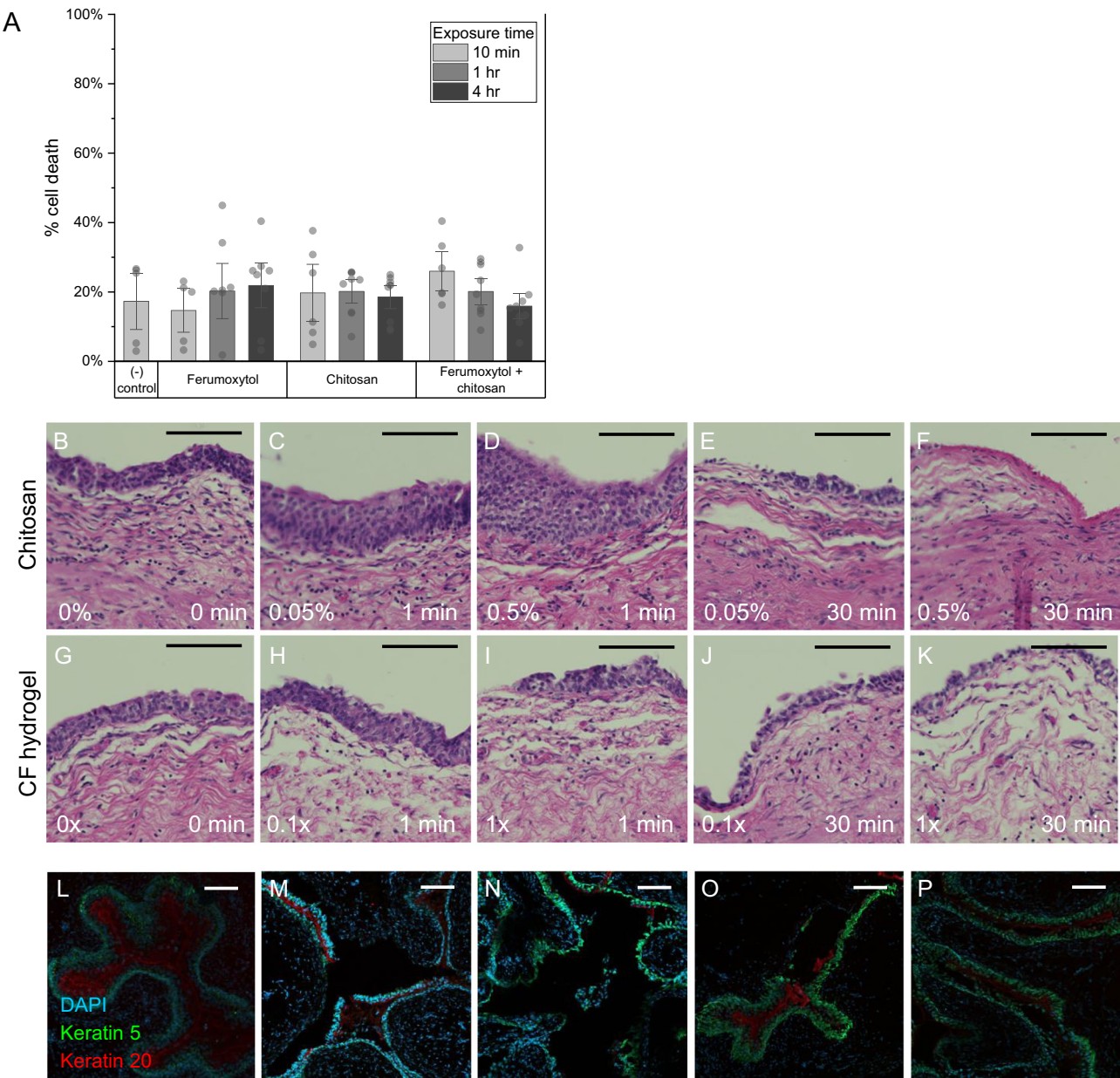

**Fig. 4 | Biocompatibility of CF hydrogel components. A** Human urothelial cancer cells were exposed to CF hydrogel components for up to 4 h. Cytotoxicity was measured by an LDH release assay. All comparisons were non-significant with $p > 0.05$ by two-tailed t-test. $n = 5$-$9$ biologically independent samples for each group. Data represents mean ± 1.5 standard error. Source data are provided as a Source Data file. **B**–**K** Histologic sections of hematoxylin & eosin-stained human urothelium incubated with hydrogel components ex vivo, taken from $n = 3$ biologically independent samples. Compared to normal urothelium (**B**), urothelium exposed to (**C**) 0.05% w/v chitosan for 1 min or (**D**) 0.5% w/v chitosan for 1 min did not show reductions in urothelial thickness. Urothelium exposed for longer durations to (**E**) 0.05% w/v chitosan for 30 min or (**F**) 0.5% w/v chitosan for 30 min showed significant exfoliation of the urothelial cell layers. Compared to normal urothelium (**G**), urothelium exposed to **H** 0.1× CF hydrogel or **I** 1× CF hydrogel for 1 min showed minimal changes in urothelial thickness. Urothelium exposed to **J** 0.1× CF hydrogel or **K** 1× CF hydrogel for 30 min showed loss of superficial cell layers, though to a lower extent compared to urothelium exposed to the same time and concentration of chitosan. **L**–**P** Fluorescent confocal microscopy images of mouse bladders exposed to hydrogel components in vivo and labeled with antibodies to DAPI (blue), keratin 5 (green), and keratin 20 (red), taken from $n = 1$ biologically independent samples for each group. Control bladders (**L**) were not exposed to any compounds. Bladders were exposed to 0.5% w/v chitosan for 15 min and examined at **M** 30 min and **N** 12 h after exposure. Immediately after exposure, there was diffuse exfoliation of the bladder urothelium, but at 12 h, basal regeneration had begun. Bladders exposed to 1× CF hydrogel (ferumoxytol and chitosan allowed to mix in the bladder) showed mild amount of exfoliation 30 min after exposure (**O**), but 12 h after exposure, the urothelium had fully recovered (**P**). Scale bar = 100 μm.

excess CF hydrogel was retrieved within 10 seconds of passive exposure to the wire (Supplementary Fig. S6).

Chitosan and the CF hydrogel may have secondary benefits in potentially reducing rates of postoperative urosepsis, as both performed comparably to the antibiotic ciprofloxacin in inhibiting the growth of two common uropathogens, *E. coli* and *P. mirabilis*. The pressurized irrigation of ureteroscopy can force bacteria colonizing the urine or kidney stones through open pyelovenous channels into the bloodstream, leading to sepsis. Despite pre-operative antibiotic prophylaxis, post-operative urosepsis after ureteroscopy occurs with an incidence of 5% and puts the patient at risk for extended hospitalization and even death[26]. Furthermore, up to 50% of stone formers had

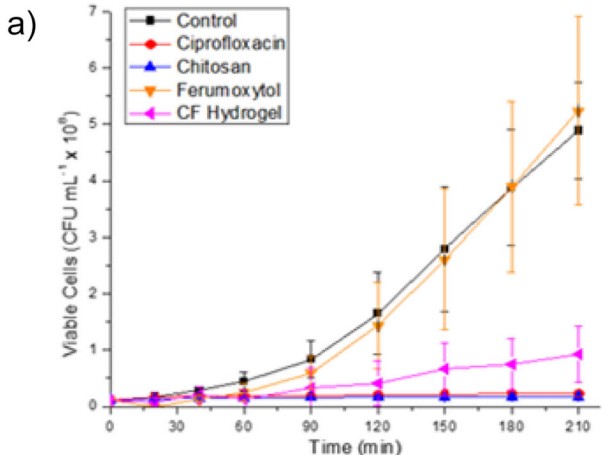

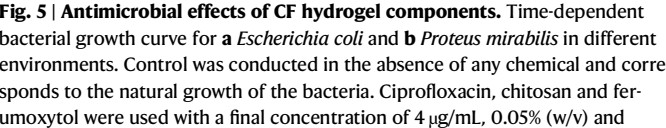

**Fig. 5 | Antimicrobial effects of CF hydrogel components.** Time-dependent bacterial growth curve for **a** *Escherichia coli* and **b** *Proteus mirabilis* in different environments. Control was conducted in the absence of any chemical and corresponds to the natural growth of the bacteria. Ciprofloxacin, chitosan and ferumoxytol were used with a final concentration of 4 μg/mL, 0.05% (w/v) and 3 mg/mL, respectively. The OD600 was recalculated to CFU/mL based on their linear relation (see Supplementary Methods). $n = 3$ independent experiments for each group. Data represents mean ± standard deviation. Source data are provided as a Source Data file.

discordant bacterial growth in cultures taken from their stones compared to their urine, increasing the risk of inadequate antibacterial prophylaxis and urosepsis[27]. Longer procedures are associated with higher rates of urosepsis, and the majority of urologists limit procedure durations to no longer than 2 h[28]. While these studies were conducted with longer durations of exposure to chitosan and CF hydrogel (>30 min) than would be expected in clinical environments under constant irrigation, the marked bactericidal activity of chitosan and the CF hydrogel is promising and warrants further study with brief (minutes-long) exposure time to bacteria both in solution and in biofilms on stone surfaces.

MagSToNE carries several potential advantages over the conventional wire basket in retrieving stone fragments. While small fragments are difficult or impossible to grasp with a basket, MagSToNE offers the possibility to retrieve small fragments by simply approximating the magnetic wire to the magnetically coated stones. Multiple magnetically coated fragments can attach to the wire and be retrieved in one pass, greatly improving the efficiency of fragment retrieval. Large stone fragments in a wire basket can become lodged in the ureter and difficult to disengage, risking ureteral injury or even avulsion. In addition, baskets can fail and even break during the course of the procedure due to the forces exerted to retrieve stone fragments[29]. In contrast, the magnetic wire is coated with a low coefficient of friction material (PTFE) which allows even stones coupled with strong normal magnetic forces to be sheared off of the wire if it experiences excessive forces within a tight ureter, which can reach 1 N[30], so a fragment that is too large to safely pass through the ureter will simply shear off of the magnetic wire. While this property of MagSToNE can help prevent injury to the ureter, it may lead to easy dislodging of captured stone fragments when withdrawing the ureteroscope and wire through a ureteral access sheath. Further optimization of magnet design, hydrogel properties, and component delivery can increase the magnetic forces and coating efficacy to improve operability while preserving the safety benefits.

We report our initial in vitro experiences with MagSToNE. While the magnetic system for stone fragment removal is promising, there remain several uncertainties that need to be studied. We intend to further study MagSToNE's performance in a benchtop ureteroscopy model to simulate the physical challenges of navigating within the limited and complex space of the kidney, as well as with irrigation and a ureteral access sheath. Particle tracing simulations suggest that small stones can be captured from several millimeters away (Supplementary Movies S1 and S2), which would allow simple removal of stone fragments that lie in recesses not easily accessible or even viewable to the surgeon[31]. Head-to-head comparisons with conventional wire baskets will also demonstrate the clinical utility of MagSToNE in reducing operative time and improving stone-free rates. Porcine studies are in progress to assess the safety and biocompatibility of the system. These studies will provide insights into the fate of the hydrogel in the urinary tract, as a potential concern is that magnetic hydrogel and magnetic hydrogel-coated stone fragments may be left behind. We postulate that the hydrogel, if not removed completely with the magnet, will be excreted by the normal urinary peristalsis and that the magnetic hydrogel will be degraded in urine, since chitosan ionic hydrogels are known to be biodegradable. Addressing these important issues will be essential towards eventual clinical translation of this promising technology.

## Methods
Our research complies with the ethical regulations set forward by the Stanford University Institutional Review Board and the Administrative Panel on Laboratory Animal Care.

### Magnetic guidewire construction
Cylindrical N50 grade neodymium iron boron (NdFeB) magnets measuring 0.75 mm in diameter by 1 mm in length, magnetized across the diameter, (Supermagnetman, Pelham, AL) were inserted into polytetrafluoroethylene (PTFE) tubing with an inner diameter (ID) of 0.76 mm and wall thickness of 12.7 μm (Zeus Inc., Orangeburg, SC). Magnets were also inserted into PTFE tubing with an ID of 0.82 mm and wall thickness of 12.7 μm (magnet to sheath surface distance of 0.05 mm), polyimide tubing with an ID of 0.89 mm and wall thickness of 25 μm (distance of 0.09 mm), and fluorinated ethylene propylene tubing with an ID of 0.91 mm and wall thickness of 200 μm (distance of 0.28 mm)

### Magnetic simulation
Simulations of different magnetic arrangements were performed with a finite element-based simulation package (COMSOL Multiphysics 5.5, COMSOL Inc., Palo Alto, CA). A detailed description of the computational modeling and relevant parameters is provided in Table 1 and the Supplementary Notes. The magnetic susceptibility value used assumes that the particles exhibit magnetic saturation. The SPIONs used in our

**Table 1 | Computational modeling and relevant parameters**

| | |
|---|---|
| Number of magnets | 4 |
| Magnet length | 1 [mm] |
| Magnet diameter | 0.75 [mm] |
| Magnet coercivity | 860,000 [A/m] |
| Magnet remnant field strength (Br) | 1.4 [T] |
| Ferumoxytol mass magnetic susceptibility | $2.2 \times 10^{-4}$ [m³/kg][33] |
| Ferumoxytol density | 7.36 [g/cm³] |
| Ferumoxytol diameter | 7 [nm] |
| $\mu_0$ (magnetic permeability of vacuum) | $4\pi \times 10^{-7}$ [T m/A] |

study saturate around 250 mT[32] and the magnetic flux density at the surface of our magnetic wire is > 0.5 T (Supplementary Fig. S1).

**Magnetic retrieval of kidney stone fragments**
Human kidney stone fragments obtained from lithotripsy (calcium phosphate (CaP), calcium oxalate monohydrate (COM), calcium oxalate dihydrate (COD), uric acid, and struvite) were separated by size. Fragments (10–15 mg by dry weight) were rehydrated with normal saline (0.9% NaCl) and incubated with ferumoxytol (AMAG Pharmaceuticals, Waltham, MA) alone or with ferumoxytol and 0.5% chitosan. Low molecular weight chitosan (50,000–190,000 Da, 75–85% deacetylated) was purchased from Sigma Aldrich and dissolved in 1% v/v acetic acid to a concentration of 0.5% w/v, pH adjusted to pH 4.5, and stored at 4 °C.

Stone fragments incubated with ferumoxytol alone were submerged in 475 μL of normal saline to simulate the aqueous environment of ureteroscopy. Twenty five microliters of ferumoxytol (30 mg iron/mL) was pipetted on top of the stones, avoiding mixing. The stones and ferumoxytol were incubated together at 37 °C for 3 min. The magnetic wire was then inserted into the solution and withdrawn. Any stone fragments which remained attached to the magnetic wire were removed and allowed to air dry for 24 h. The dried fragments were weighed, and the capture efficiency was determined as the dry weight of the captured fragments divided by the starting stone dry weight.

Stone fragments incubated with ferumoxytol, and chitosan were submerged in 425 μL of normal saline at room temperature. Twenty five microliters of 30 mg/mL ferumoxytol was pipetted on top of the stones, avoiding mixing. Immediately after, 50 μL of 0.5% w/v chitosan was pipetted into the layer of ferumoxytol and stone and agitated with the tip of the pipet to form a chitosan-ferumoxytol (CF) hydrogel. Immediately after, the magnetic wire was introduced to retrieve the hydrogel-coated stones. Captured fragments were similarly dried and weighed. In contrast to excess ferumoxytol alone, the excess CF hydrogel was also attracted to the magnetic wire. When calculating capture efficiencies for stones captured with CF hydrogel, 0.9 mg was subtracted from the dry weight to account for the maximum dry weight of the CF hydrogel. $n \geq 3$ for all groups.

Magnetic retrieval of kidney stone fragments with larger superparamagnetic nanoparticles and beads. Stone fragments were hydrated by mixing with 50 μL of normal saline. The excess fluid was removed. Twenty five microliters of iron oxide nanoparticles (7 nm: ferumoxytol (30 mg iron/mL) (AMAG Pharmaceuticals, Waltham, MA); 50–100 nm: SuperMag Carboxyl Beads (10 mg/mL) (Ocean NanoTech, San Diego, CA)) or beads (1 μm: DynaBeads MyOne Carboxylic Acid (10 mg/mL) (Thermo Fisher Scientific, Waltham, MA); 3–4.5 μm MonoMag Carboxyl Beads (30 mg/mL) (Ocean NanoTech, San Diego, CA)) at their stock solution was pipetted on top of the stones. The stones were then incubated at 37 °C for 3 min. Four hundred fifty microliters of normal saline was added to the solution. The magnetic wire was then inserted into the solution and withdrawn. Any stone fragments which remained attached to the magnetic wire were removed and allowed to air dry for 24 h. The dried fragments were weighed, and the capture efficiency was determined as the dry weight of the captured fragments divided by the starting stone dry weight.

**Scanning electron microscopy**
Stone fragments were captured as described above with either ferumoxytol or CF hydrogel. Dried fragments were mounted with carbon tape onto stainless steel pin studs (Ted Pella Inc., Redding, CA). Samples were coated with Au/Pd (60:40 ratio) or with carbon (15 micron thickness) for energy-dispersive X-ray analysis (EDX), and observed with an Apreo S LoVAC SEM (Thermo Fisher Scientific, Waltham, MA) and Quantax XFlash 6 system (Bruker, Billerica, MA).

**Cell culture**
Human T24 bladder transitional cell carcinoma cells (ATCC, Manassas, VA, Cat # HTB-4) were grown in T-75 flasks in the presence of DMEM media supplemented with 10% fetal bovine serum (FBS) and 1% penicillin-streptomycin. Cells were maintained at 37 °C in a in a humidified atmosphere of 5% CO$_2$ (v/v) and trypsinized when they reached 80% confluence by removing the old medium and rinsing with 1× PBS, pH 7.4 (Thermo Fisher) and incubating with TrypLE Express (Thermo Fisher) for 5 min at 37 °C. Cells were diluted with DMEM + 10% FBS media, spun down at $150 \times g$ for 5 min, and then resuspended in their respective media. Viable cells were quantified by mixing 20 μL of cells 1:1 with Trypan Blue solution (Thermo Fisher) and counting manually with a hemocytometer.

96-well flat clear-bottom plates were inoculated with 10,000 cells each. For cytotoxicity and viability studies, cells in each well were incubated in a total volume of 100 μL of cell medium with either 10 μL PBS (negative control), 10 uL 10x lysis solution (Promega), 5 μL 30 mg/mL ferumoxytol (final concentration 1.5 mg/mL), 10 μL 0.5% w/v LMW chitosan pH 4.5 (final concentration 0.05% w/v), or 5 μL ferumoxytol mixed with 10 μL chitosan. Cells were incubated at 37 °C for 4 h with the negative control, 45 min with the positive control, and for 1 h or 4 h with the hydrogel components. Blank solutions were also created using cell medium with PBS, ferumoxytol, and ferumoxytol with chitosan, to account for the absorbance at 490 nm of ferumoxytol.

At the conclusion of the incubation period, 50 μL of solution from each well was transferred to a fresh 96-well flat clear-bottom plate for the CytoTox 96 Assay (Promega, Madison, WI), an LDH release assay. Fifty microliters of CytoTox 96 Reagent (Promega) was added to each well. The plate was covered with an opaque box and incubated for 30 min at room temperature. Fifty microliters of Stop Solution (Promega) was added to each well. Large bubbles were popped with a syringe needle and the absorbance at 490 nm was recorded with an EL800 spectrophotometer (BioTek, Winooski, VT). The percent cytotoxicity was calculated as the OD$_{490}$ of the experimental wells divided by the OD$_{490}$ of the positive control wells. $n \geq 5$ for all groups.

**Ex vivo human urothelium studies**
All studies were done with prior approval from the Stanford University and Palo Alto Veterans Affairs Healthcare Institutional Review Boards. Informed consent was provided by the patients from whom the human samples were obtained from. Human urothelium was obtained from fresh nephrectomy specimens. Nephrectomy specimens with suspicion for urothelial carcinoma were not used. The ureter and renal pelvis were excised and placed on Telfa moistened with normal saline, and flattened with a circular grid to create individual wells. Each area of urothelium was exposed to either PBS (negative control), ferumoxytol, chitosan, or the CF hydrogel, and incubated at 37 °C for 1 or 30 min. The urothelium was then gently rinsed with normal saline and fixed in formalin for 24 h. Tissue samples were embedded in paraffin and stained with hematoxylin & eosin or Prussian blue (Histo-Tec Laboratory, Hayward, CA). Samples were observed with a Keyence BZ-X810

optical microscope (Keyence, Osaka, Japan). Experiments were performed in triplicate.

## Mouse studies

All animal procedures were done with prior approval from the Administrative Panel on Laboratory Animal Care at Stanford University. Female wild type mice between 2–6 months of age were purchased from Jackson Laboratory. Mice were housed with a 12-h light dark cycle with room temperature 71–72 F and humidity 29–40%, and fed a diet of Teklad Global 18% Protein Extruded Rodent Diet (Inotiv, Chicago, IL). Mice were anesthetized with isoflurane and placed supine. The urethra was catheterized with polyethylene tubing (BD Intramedic PE Tubing PE10) fitted onto a 30 G needle. The bladder was emptied by aspirating urine through the catheter. Fifty microliters of PBS, 0.5% w/v chitosan, 0.05% w/v chitosan, or 1× CF hydrogel (20 μL 30 mg/mL ferumoxytol immediately followed by 30 μL of 0.5% w/v chitosan) was instilled into the bladder. The mice were left anesthetized for 15 min to prevent immediate voiding of bladder contents. After 30 min or 12 h, mice were euthanized by placing in a $CO_2$ chamber until breathing and movement had ceased for 1 min, followed by cervical dislocation. The bladder was harvested and placed into Optimal Cutting Temperature medium (OCT; Fisher Scientific), and flash frozen on dry ice and stored at −80 °C.

Frozen sections were taken at 14 μm thickness using a cryostat (Leica) and mounted onto slides, and then fixed with ice cold 4% paraformaldehyde/1× PBS for 25 min, washed two times in 1× PBS, and permeabilized in 0.25% Triton X-100/1XPBS for 15 min at room temperature. Sections were then blocked in 5% normal goat serum (NGS; Jackson ImmunoResearch) for 30 min at room temperature in a humidified chamber. Primary antibody incubations were performed in blocking solution overnight in a humidified chamber at 4 °C, followed by three 15 min washes in 0.25% Triton X-100/1XPBS. Primary antibodies used were chicken anti-Krt5 (1:1000; 905901, BioLegend) and guinea pig anti-Krt20 (1:500; GP-K20, ProGen). Secondary antibodies (Alexa Fluor 633 anti-chicken, or 488 anti-guinea pig, 1:500; Molecular Probes) were diluted at 1:1000 in blocking solution containing 1 μg/mL 4′,6-diamidino-2-phenylindole (DAPI) to visualize DNA, applied to samples, and incubated at room temperature for 2 h in a humidified chamber. Samples were observed with an inverted laser-scanning confocal microscope (Zeiss LSM 800) using a 10X lens (Zeiss Plan-Apochromat 20×/0.8 M27) and Zen 2.1 software, blue edition (Zeiss).

## Bacteria

Human uropathogens (*E. coli*, *P. mirabilis*) were previously isolated from urine culture and frozen at −80 °C. Bacteria were inoculated from frozen stocks onto Mueller Hinton (MH) agar plates and grown statically at 37 °C overnight. A single colony was used to inoculate a MH broth solution. Once bacteria were in the log phase of growth, the solution was diluted to an OD600 of 0.02, corresponding to approximately $10^7$ colony-forming units (CFU)/mL.

96-well flat clear-bottom tissue-culture treated plates were inoculated with 70 μL of bacterial suspension ($10^7$ CFU/mL) and control (MH broth) or compound to a final volume of 200 μL and incubated at 37 °C on a horizontal shaker. The final concentrations of reagents used were as follows: ciprofloxacin 4 μg/mL (Acros Organics), chitosan 0.05% w/v, and ferumoxytol 3 mg/mL. For experiments involving the CF hydrogel, the precursors were mixed prior to introduction of the bacterial solution. The OD600 was recorded at 30 min intervals over a 4-h period to quantify bacterial growth.

After 4 h of incubation, bacterial suspensions from each group were serially diluted and isolated via the spread plate technique onto MH agar plates. Plates were incubated at 37 °C and counted after 24 h to calculate the CFU/mL of the original sample after 4 h of growth. Each study was performed in triplicate.

## Hydrogel clearance

To evaluate hydrogel clearance with the magnet, 25 μL of 30 mg/mL ferumoxytol, with or without 50 μL of 0.5% w/v chitosan, was mixed in 500 μL total volume of 0.9% saline in a 1.5 mL Eppendorf tube (final ferumoxytol concentration 1.5 mg/mL). A 2 cm length of magnetic wire was placed into the tube for the given cumulative time without further agitation or mixing. At each time interval, the wire was removed and wiped clean. The mixture was vortexed and the A475nm (corresponding to the absorbance peak of ferumoxytol) of the solution was measured using a Nanodrop 2000. Measurements were taken in triplicate. A calibration curve of A475 versus concentration was created for ferumoxytol to determine the percentage of ferumoxytol remaining in solution.

## Blood clearance

Human blood (15 g/dL hemoglobin) was diluted 50-fold in 0.9% saline to represent a typical concentration of blood which would begin to obscure vision during ureteroscopy. Twenty five microliters of 30 mg/mL ferumoxytol, with or without 50 μL of 0.5% w/v chitosan, was mixed with the diluted blood in 500 μL total volume in a 1.5 mL Eppendorf tube. A 2 cm length of magnetic wire was placed into the tube for the given cumulative time without further agitation or mixing. At each time interval, the wire was removed and wiped clean. The mixture was vortexed and the A540nm of the solution (corresponding to the absorbance peak of hemoglobin) was measured using a Nanodrop 2000. Measurements were taken in triplicate. A calibration curve of A540 versus concentration was created for both blood and blood with ferumoxytol to determine the amount/percentage of hemoglobin remaining in solution.

## Particle tracing simulation

The COMSOL simulation described in the main text was used to create particle tracing simulations of stone fragments coated in ferumoxytol to determine the distance from the wire for magnetic capture. A 1 mm stone fragment was estimated to be coated with $2 \times 10^{12}$ ferumoxytol particles in a hydrogel, by calculating the maximum number of 7 nm ferumoxytol particles that could cover the surface of a spherical 1 mm stone fragment and multiplying this by a factor of 100−a conservative estimate, given that the hydrogel layer of 7 nm particles on a stone can be many microns thick. The Particle Tracing for Fluid Flow module was used to simulate the magnetophoretic force on a coated stone fragment and the opposing drag force in water, at distances up to 4 mm away from the magnet.

## Reporting summary

Further information on research design is available in the Nature Portfolio Reporting Summary linked to this article.

## Data availability

The source data generated in this study have been deposited in a public repository at https://doi.org/10.17605/OSF.IO/A843D. Source data are provided with this paper.

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

## Acknowledgements

Funding was received from NIH R21 DK131776 (T.J.G., J.C.L., S.X.W., S.C.), NIH R25 DK122957 (G.H.H., J.C.L.), Stanford Coulter Translational Seed Grant (T.J.G., J.C.L., S.X.W., S.C., K.S.), and the American Urologic Association/Urology Care Foundation Resident Research Award (T.J.G., J.C.L., S.C., K.S.). Part of this work was performed at the Stanford Nano Shared Facilities (SNSF), supported by the National Science Foundation under award ECCS-2026822.

## Author contributions

T.J.G. and J.C.L. designed the study. K.E.M., T.C.C., S.C., K.S., S.X.W., and J.C.L gave conceptual advice and assisted in study design. T.J.G., D.M.R., G.H.H., and K.P. performed experiments and analyzed data. H.L., K.J., and T.C.C. provided stone specimens and nephrectomy specimens. T.J.G. and D.M.R. wrote the manuscript.

## Competing interests

Authors T.J.G., S.C., K.S., S.X.W., and J.C.L. are inventors on patent US 2022/0160450, filed November 11, 2021, patent pending. This patent covers the MagSToNE technology described in the article. The remaining authors declare no other competing interests.
