## [Peer Review File · Nature Communications]

A magnetic hydrogel for the efficient retrieval of kidney stone fragments during ureteroscopyREVIEWER COMMENTS

Reviewer #2 (Remarks to the Author):

The idea of preparing a magnetic gel based on biopolymer (chitosan) and FDA-approved SPION (ferumoxytol) for the removal of kidney stones seems to be innovative and interesting. The obtained results are scientifically meaningful and practically important. However, revision is needed.

1. Explain why the 7 nm SPION was finally chosen for the gel formation. From the data presented in Fig. S1 b) it seems that the larger ones seem more reasonable.
2. The structure of the gel should be further investigated and discussed. E.g. FT-IR spectroscopy, SEM microscopy (after lyophilization of the sample) can be useful.
3. Interaction of SPINs with kidney stones via carboxylic groups are clear. However, the interaction of chitosan with kidney stones should be investigated and explained.
4. References to figures in the supplement require correction.

Reviewer #3 (Remarks to the Author):

The novelties introduced in this paper, in view of prior efforts, notably include (a) radial rather than axial magnetization of the retrieval tool, and (b) use of an engineered hydrogel to increase SPION loading on the kidney stones -- both noteworthy and interesting ideas.

In point of the former, the authors explain how radial magnetization increases available surface area and improves "per pass" stone burden retrieval, acknowledging the penalty endured from the sheathing necessary to sustain the assembly.

Apparently missing, however, is an assessment of the target distance at which the authors' retrieval tool is effective. This consideration was a key focus of prior efforts, since offending stone remnants are liable to lie in recesses not swept by the retrieval tool, or even viewable to the surgeon.

It is uncontroversial and well-established that on-contact magnetic retrieval of SPION-loaded stones is possible. While improvements in magnetostatic circuit design are worthy contributions, a discussion on distance effectiveness is relevant.

This was a light review on my part due to time constraints, so I tender my apologies if I missed the authors' intent in this regard.

We thank the reviewers for their overall enthusiasm for our manuscript and their insightful comments. Below please find our detailed response to their comments.

Reviewer 2

Overall Comment:

The idea of preparing a magnetic gel based on biopolymer (chitosan) and FDA-approved SPION (ferumoxytol) for the removal of kidney stones seems to be innovative and interesting. The obtained results are scientifically meaningful and practically important. However, revision is needed.

Comment 1: *Explain why the 7 nm SPION was finally chosen for the gel formation. From the data presented in Fig. S1 b) it seems that the larger ones seem more reasonable*

Response 1:

When comparing particles alone, which form a monolayer coating on the stone surface, the larger magnetic particles with larger magnetic moments yielded higher capture efficiency (Fig. S1). However, reliable capture of stone fragments > 1 mm in size was not possible without a thicker hydrogel layer coating to increase the number of particles per fragment. The larger particles (100s of nm to micron in size) did not easily cross-link with our polymer to form a gel, likely due to an unfavorable surface-to-volume ratio, while the 7 nm SPION formed a much more robust gel. The 7 nm SPION ferumoxytol was ultimately chosen for a) its ease of use as a cross-linking agent to form a hydrogel, and b) its potentially streamlined path towards clinical translation, given ferumoxytol is an FDA-approved drug. The resulting magnetized hydrogel was successful in capturing all stone types < 4 mm (Fig. 2). We have further clarified this important point in the Results and Discussion sections of the revised manuscript.

Comment 2: *The structure of the gel should be further investigated and discussed. E.g. FT-IR spectroscopy, SEM microscopy (after lyophilization of the sample) can be useful.*

Response 2:

We agree with the reviewer and have now included a new figure in the supplementary information of the revised manuscript that includes SEM and FT-IR of the lyophilized hydrogel. A short description of them was included in the results section.

Comment 3: Interaction of SPINs with kidney stones via carboxylic groups are clear. However, the interaction of chitosan with kidney stones should be investigated and explained.

Response 3:

Chitosan is used at a pH in which it is positively charged. This biopolymer was selected to cross-link to the negatively charged nanoparticles already coating the stone and form a thick hydrogel coating that contains more magnetic nanoparticles, not to primarily bind to the kidney stone surface. Chitosan likely interacts with the stone by adsorption typical of rough surfaces. The interaction of chitosan and kidney stones is of scientific interest, but it is not necessary for the intended application of the hydrogel. We have provided additional clarification in the Results to highlight this important point.

Comment 4: References to figures in the supplement require correction.

Response 4:

References and numbering in the text have been corrected to be in the proper order.

Reviewer 3

Comment 1: *The novelties introduced in this paper, in view of prior efforts, notably include (a) radial rather than axial magnetization of the retrieval tool, and (b) use of an engineered hydrogel to increase SPION loading on the kidney stones – both noteworthy and interesting ideas.*

In point of the former, the authors explain how radial magnetization increases available surface area and improves “per pass” stone burden retrieval, acknowledging the penalty endured from the sheathing necessary to sustain the assembly.

Response 1:

We thank the reviewer for recognizing the innovative features of our study.

Comment 2: *Apparently missing, however, is an assessment of the target distance at which the authors' retrieval tool is effective. This consideration was a key focus of prior efforts, since offending stone remnants are liable to lie in recesses not swept by the retrieval tool, or even viewable to the surgeon.*

It is uncontroversial and well-established that on-contact magnetic retrieval of SPION-loaded stones is possible. While improvements in magnetostatic circuit design are worthy contributions, a discussion on distance effectiveness is relevant.

Response 2:

We concur with the reviewer on the importance of the target distance from our magnetic system. Prior works by Fernandez et al (*J. Endourol.* 26, 1227–1230 (2012)) have shown theoretical capture distances of a few millimeters decreasing logarithmically with stone size, since the weight of the stone fragment grows in cubic proportion to radius; whereas the surface magnetization of a stone coated with particles grows only in square proportion to radius. We have included this prior work as Reference #31 in the revised manuscript.

A key difference in our work is that the thicker hydrogel layer of magnetic particles significantly increases the surface magnetization, and the magnetic gradient generated by the retrieval tool is larger, thereby allowing for theoretical capture from further distances. We have conducted particle tracing simulations to

assess the distance from which stones can be captured by the magnet and included a Supplementary Video S1, showing that 1 mm diameter coated stones can be captured from up to 4 mm away.

Practically, the magnetic tool is a flexible wire that can be steered using the flexible ureteroscope to reach any area that can be visualized by the surgeon, and beyond, as the surgeon can 'blindly' advance the atraumatic wire while deflecting it with the flexible ureteroscope in order to access around corners and capture stones from recesses. We anticipate that the nimbleness of the design combined with a magnetic reach of several millimeters should allow for capture of stone fragments from most areas.

Lastly, the capture distance depends on the orientation of the magnetic wire and hydrogel coating thickness of the stone. Ongoing pilot experiments in testing the proposed magnetic system in clinically relevant ex vivo and in vivo models show that the actual capture distance is similar to that shown in our simulation. These additional experiments will help understand better the minimum distance and will be reported in our future publication.

The following changes were added in the manuscript.

A new discussion was included in the conclusions section:

Particle tracing simulations suggest that small stones can be captured from a distance of several millimeters away (Video S1), which would allow simple removal of stone fragments that lie in recesses not easily accessible or even viewable to the surgeon.

New supplementary methods and a video were included in the supplementary section:

Particle tracing simulation. The COMSOL simulation described in the main text was used to create particle tracing simulations of stone fragments coated in ferumoxytol to determine the distance from the wire for magnetic capture. A 1 mm stone fragment was estimated to be coated with 2×10^{12} ferumoxytol particles in a hydrogel, by calculating the maximum number of 7 nm ferumoxytol particles that could cover the surface of a spherical 1 mm stone fragment and multiplying this by a factor of 100 – a conservative estimate, given that the hydrogel layer of 7 nm particles on a stone can be many microns thick. The Particle Tracing for Fluid Flow module was used to simulate the magnetophoretic force on a coated stone fragment and the opposing drag force in water, at distances up to 4 mm away from the magnet.

Supplemental Videos – see attached file

Video S1. A cross-section of the magnetic wire is shown with radial magnetization in the z direction. The magnetic gradient exerted by the wire is shown in the red-blue logarithmic color scale, showing that the gradients are stronger in the z axis compared to the y axis. Gravitational force is not simulated, as the particles (1 mm diameter magnetically labeled stone fragments) are assumed to be settled on the floor of the kidney, thus the only active forces are magnetophoresis and drag. Particle motion is simulated over 1 second. Particles along the z-axis are attracted to the wire nearly instantaneously (< 0.1 second), from a distance of up to 4 mm, while particles off of the z-axis are attracted more slowly from a distance of 1 mm.

REVIEWER COMMENTS

Reviewer #2 (Remarks to the Author):

I recommend publication of the revised ms.

Reviewer #4 (Remarks to the Author):

The added sentence about capture distance and supporting video is helpful but misleading (lines 260-262).

The capture distance is critical to the clinical utility of this technology.

The video shows only stone fragments of 1 mm in size being attracted/moved.

What of larger clinically significant fragments 2-4 mm in size?

They do not seem to move?

And though the authors state the scope can fragments, urologic surgeons know that though once could see a fragment, cannot always get a wire/basket to actually touch/get close to a residual fragment.

This thus raises issue of retained fragments 2-4 mm that cannot be attracted staying in patient with chitosan coating and as raised by reviewer, implications on stone growth not known.

By introducing a foreign substance that could stay and not be washed out, impact as stone nidus is unknown.

We thank the reviewers for their overall enthusiasm for our manuscript and their insightful comments. Below please find our detailed response to their comments.

Reviewer 2

Comment:

I recommend publication of the revised ms.

Response:

We thank the reviewer for the recommendation.

Reviewer 4

Comment 1: *The added sentence about capture distance and supporting video is helpful but misleading (lines 260-262). The capture distance is critical to the clinical utility of this technology. The video shows only stone fragments of 1 mm in size being attracted/moved. What of larger clinically significant fragments 2-4 mm in size? They do not seem to move?*

Response 1:

We appreciate the reviewer's comments and would like to provide additional clarification. We concur that the capture distance is critical to the clinical utility of this technology. The stones simulated in Video S1 in NCOMMS-22-47731A are all 1 mm in size. The numbers along the vertical and horizontal axes in the original video is referring to the stone's distance away from the magnet, rather than the stone diameter. The cross-section of the magnet is depicted as a red/blue circle centered at the origin. To improve clarity, we have now provided a **revised Video S1** (screenshot below) that clarified that all stones are 1 mm in size and a radial depiction of distance from the magnet in mm.

Stones up to 3.5 mm away from the magnet experience a net attractive magnetophoretic force, and once a stone is within 2 mm of the magnet it is nearly instantaneously attracted to the magnet. The caption of the Video S1 has also been revised to improve the clarity. We have also conducted simulations for stones 3 mm in size, which showed similar results for capture distance (data available upon request). These simulation experiments are consistent with our experimental data shown in **Figure 3a** that demonstrates successful capturing of stone fragments of varying sizes (< 1 mm, 1-2 mm, and 2-4 mm) with the chitosan-ferumoxytol hydrogel.

Comment 2: *And though the authors state the scope can fragments, urologic surgeons know that though once could see a fragment, cannot always get a wire/basket to actually touch/get close to a residual fragment.*

Response 2:

We agree that it can be difficult to precisely capture a fragment within the tines of a basket, particularly in less accessible parts of the renal collecting system. It is generally easier to contact the fragment or to get within a few millimeters of the stone fragment with a wire than capture with a basket. This is one of the main advantages of our approach, as it removes the complexities of having to open and close a basket around the fragment.

Comment 3: *This thus raises issue of retained fragments 2-4 mm that cannot be attracted staying in patient with chitosan coating and as raised by reviewer, implications on stone growth not known. By introducing a foreign substance that could stay and not be washed out, impact as stone nidus is unknown.*

Response 3:

As mentioned in response to comment 1, we can capture 2-4mm stone fragments. We agree with the reviewer's concerns that residual gel may serve as a nidus for stone growth. Chitosan is known to be biodegradable and our data suggests that the majority of the magnetized hydrogel is removed with the stones. Our ongoing animal studies are designed to address this issue.

Supplemental Videos – see attached file

Video S1. A cross-section of the magnetic wire is shown with radial magnetization in the z direction. The magnetic gradient exerted by the wire is shown in the red-blue logarithmic color scale, showing that the gradients are stronger in the z axis compared to the y axis. Gravitational force is not simulated, as the particles (1 mm diameter magnetically labeled stone fragments) are assumed to be settled on the floor of the kidney, thus the only active forces are magnetophoresis and drag. Particle motion is simulated over 1 second. Particles along the z-axis are attracted to the wire nearly instantaneously (< 0.1 second), from a distance of up to 4 mm, while particles off of the z-axis are attracted more slowly from a distance of 1 mm.

We thank the reviewers for their overall enthusiasm for our manuscript NCOMMS-22-47731 and their insightful comments. Below please find our detailed response to their comments.

Reviewers Comment

Comment 1:

Whilst we accept that in vivo demonstration of this system will come in a later manuscript and your discussions around safety and demonstration of clearance are strong, we request that you add a clear caveat that clinical/practical utility remains to be determined with capture distance and complexity of intrarenal surgery being unknowns.

Response 1:

We included a statement that emphasize the importance of stone capture distance for clinical translation as well as our current efforts to investigate the practical/clinical utility of the approach in 3D kidney models that simulate the complexity of intrarenal surgeries.

Comment 2:

We also request that you add potential limitations for this system regarding issues such as stone growth and residual coated stones being uncaptured etc.

Response 2:

We included a statement of the potential concerns of the hydrogel left behind in the kidney as a nidus for stone formation.

Comment 3:

Finally, we also request that you include the simulations for 3mm stones in the manuscript.

Response 3:

Simulation for 3 mm magnetized stones have been included in the supplementary information (Video S2)